# Body Height of MPS I and II Patients after Hematopoietic Stem Cell Transplantation: The Impact of Dermatan Sulphate

**DOI:** 10.3390/diagnostics14171956

**Published:** 2024-09-04

**Authors:** Patryk Lipiński, Agnieszka Różdżyńska-Świątkowska, Agnieszka Ługowska, Jolanta Marucha, Katarzyna Drabko, Anna Tylki-Szymańska

**Affiliations:** 1Institute of Clinical Sciences, Maria Skłodowska-Curie Medical Academy, 03-411 Warsaw, Poland; 2Anthropology Laboratory, The Children’s Memorial Health Institute, 04-730 Warsaw, Poland; a.rozdzynska-swiatkowska@ipczd.pl; 3Department of Genetics, Institute of Psychiatry and Neurology, Al. Sobieskiego 9, 02-957 Warsaw, Poland; alugipin@yahoo.com; 4Department of Pediatrics, Nutrition and Metabolic Diseases, The Children’s Memorial Health Institute, 04-730 Warsaw, Poland; j.marucha@ipczd.pl; 5 Department of Pediatric Hematology, Oncology and Transplantology, Medical University of Lublin, 20-093 Lublin, Poland; katarzynadrabko@umlub.pl

**Keywords:** height, hematopoietic stem cell transplantation, mucopolysaccharidosis, children

## Abstract

Introduction: Hematopoietic stem cell transplantation (HSCT) comprises one of the two main treatment regimens for patients with mucopolysaccharidoses (MPS). There is a scarcity of literature concerning the process of growth in children with Mucopolysaccharidosis type I (MPS I) and Mucopolysaccharidosis type I (MPS II) after HSCT. The aim of this manuscript was to evaluate the therapeutic effect of HSCT on the heights of patients with MPS I and MPS II. Material and methods: It was an observational, single-center study on patients with MPS I and II treated with HSCT. Results: 6 MPS patients, including 4 MPS I and 2 MPS II, underwent HSCT at a median age of 2 years. All patients are alive to date, with a median age of 7.7 years (range 5.5–12 years) at the last follow-up. In both (MPS I and MPS II) groups of patients treated with HSCT, the growth rate was higher than in untreated patients and was found to be in line with the population norm. In both MPS I and MPS II patients who were treated with HSCT, normalization of urinary GAG excretion was observed. Additionally, no bands of DS and HS in GAG electrophoresis were visible. Conclusions: Both MPS I and MPS II patients presented height gain after HSCT compared to the curves of untreated patients. The absence of dermatan sulphate after HSCT could lead to normal growth in bone length.

## 1. Introduction

Mucopolysaccharidoses (MPS) are a group of inherited metabolic diseases related to deficiencies of lysosomal enzymes responsible for glycosaminoglycans (GAGs) catabolism [1,2]. GAGs consist of five types of disaccharides: heparan sulphate (HS), dermatan sulfate (DS), chondroitin sulfate (CS), keratan sulfate (KS), and hyaluronan [3]. Depending on the impaired enzyme type, the accumulated GAGs differ, leading to specific abnormalities [2,3]. DS is primarily found in the skin, cartilage, bones, cornea, cardiac valves, and other connective tissues. Its accumulation has a direct detrimental effect on the cartilage, in which damage to the growth plates undoubtedly harm the growth and volume of bone [3,4]. Abnormal HS levels in the central nervous system (CNS) have been found to be responsible for dysregulation of neuronal differentiation, growth, and neurotransmission [5].

Mucopolysaccharidosis type I (MPS I) is an autosomal recessive lysosomal storage disease (LSD) associated with deficient activity of α-L-iduronidase (IDUA; EC 3.2.1.76) [6]. According to its clinical severity and neurodevelopmental outcome, three forms of MPS I are distinguished, namely, Hurler (MPS I-H; # 607014), Hurler–Scheie (MPS I-HS; # 607015), and Scheie syndromes (MPS I-S; # 607016) [6]. Mucopolysaccharidosis type II (MPS II, Hunter syndrome; # 309900) is an X-linked LSD caused by a deficiency of iduronate 2-sulfatase (I2S, EC 3.1.6.13) [7]. Patients with MPS II are currently categorized as having either an attenuated (no central nervous system (CNS) impairment) or severe (CNS impairment) phenotype, with the latter being the more prevalent. DS and HS are found as primary storage materials in both MPS I and MPS II (neuronopathic forms).

Two main treatment regimens for MPS patients are available in practice: hematopoietic stem cell transplantation (HSCT) and enzyme replacement therapy (ERT). MPS I was the first to be treated with ERT (Aldurazyme), which has been available since 2003 [6,8]. In 2006, ERT with recombinant human I2S (Elaprase) was approved for the treatment of MPS II [7,9]. ERT does not cross the blood–brain barrier (BBB), and this fact especially concerns patients with severe (neuronopathic) forms of MPS. The first bone marrow transplantation in MPS was performed in a 1-year-old boy with MPS IH in 1981 [10]. HSCT is now reserved for patients below the age of 2 years (before severe cognitive impairment) with the most severe type of MPS and constitutes the gold standard treatment for MPS IH [11]. Its efficacy depends on the patient’s age and disease stage at the time of the procedure, the type of MPS, the type of donor, and the preparative regimen. While survival rates and short-term clinical outcomes are known to be improved by HSCT, there is a scarcity of literature concerning the process of growth in children with both MPS I and MPS II after HSCT.

The aim of this manuscript was to evaluate the therapeutic effect of HSCT on the height of patients with MPS I and MPS II. A comparison with untreated patients was also provided.

## 2. Material and Methods

### 2.1. Anthropometric Measurements

Anthropometric measurements were taken according to a standard technique and included body length/height. Until the age of 3 years, the body lengths of the children were measured in supine position using a liberometer (accuracy to 1 mm). The same measurements in older children were performed for standing height using a stadiometer (accuracy to 1 mm).

The study group patients did not present with skeletal deformities that would affect measurements. Also, the comparable measurements in a lying position along the curves of the body with a centimeter confirmed the above-mentioned information.

### 2.2. GAG Electrophoresis

GAGs were isolated from urine sediment by precipitation with Cetylpyridinium chloride (CPC) and electrophoresed on the acetyl acetate according to the method of Hopwood [12].

## 3. Patient Characteristics

### 3.1. Overall Characteristics

The median age at MPS diagnosis was 14 months (range 13–16 months). In all patients, diagnosis was confirmed by increased urinary GAGs and electrophoresis, as well as deficiency of an adequate enzyme (IDUA or I2S activity) in leukocytes/dried blood spots. Mutation analysis revealed all patients with MPS I (Pt 1–4) to be homozygous for p.Q70X variant. Patients 5 and 6 were found to be hemizygous for a pathogenic deletion encompassing exon 8 and exons 1 to 8 of the *IDS* gene, respectively.

Six MPS patients, including four MPS I and two MPS II, underwent HSCT at a median age of 2 years (range 18 months–2 years and 6 months). All HSCT grafts were bone marrow transplants (BMT). Within the first year post-HSCT, full donor chimerism (100%) was achieved in all but one patient (Pt 2 with MPS I). In Patient 2, the chimerism decreased below 50% from 6 months years post HSCT. He suffered from severe hemolytic anemia (diagnosed 6 months after HSCT) and was treated with corticosteroids, cyclosporine, and rituximab (4 doses). Finally, he underwent 2nd HSCT one year after.

All patients are alive to date, with a median age of 7.7 years (range 5.5–12 years) at the last follow-up. The median follow-up time post-HSCT was 5.5 years (range 2.5–9.5 years). 

### 3.2. Clinical Outcome

Thoracolumbar spine kyphosis (MPS I) as well as hip dysplasia (MPS I and II) were observed in all patients at the time of diagnosis. There was a tendency to progression of both spine kyphosis and hip dysplasia observed, despite HSCT. However, only one patient (Pt 6) required surgical intervention (due to hip luxation). At diagnosis, the biggest restriction in passive range of movement (ROM) in shoulder abduction and flexion was observed. After HSCT, there was stabilization or slight improvement. 

No patient had hydrocephalus. Narrowing of the spinal canal was observed in one patient (Pt 1) starting at the age of 5.5 years.

No patient presented with carpal tunnel syndrome before HSCT. Two patients (Pt 2 and 4) developed carpal tunnel syndrome, at 5 and 12 years, respectively.

Corneal clouding was observed in all patients at the time of diagnosis of MPS I. Despite HSCT, corneal clouding worsened in all patients.

All patients but one (Pt 1) underwent adenectomy with tympanostomy tube fitting before HSCT. The latter was qualified for these procedures after HSCT at 5.5 years of age.

Bilateral sensorineural hearing impairment was observed in four patients and was assessed as mild or moderate (40–55 dB) at the time of diagnosis. Long-term hearing appeared to improve in two patients (unilateral mild hearing loss) and to stabilize in two other patients after HSCT.

Mitral valve thickening was observed in all patients before HSCT and remained stable after HSCT.

All the patients presented with normal intellectual development after HSCT.

Detailed clinical characteristics of the study patients are provided in Table 1.

### 3.3. Anthropometric Phenotype

Patients with MPS I and MPS II after HSCT had normal body proportions, including the length of the trunk and lower limbs, and normal head circumferences, like healthy children. Narrow shoulders and chests, as in untreated MPS patients, were also observed in MPS I after HSCT.

In both (MPS I and MPS II) groups of patients treated with HSCT, the growth rates were higher than in untreated patients and were found to be in line with the population norm, see Figure 1 and Figure 2. 

Individual patients’ growth charts were provided in Appendix A. All the patients presented with height gain after HSCT.

### 3.4. Urinary GAG Electrophoresis

In both MPS I and MPS II patients who were treated with HSCT, normalization of urinary GAG excretion was observed when comparing the results of the semi-quantitative test with CPC (Table 1). Additionally, in both (MPS I and MPS II) groups of patients after HSCT, no bands of DS or HS in GAG electrophoresis were visible (Figure 3 and Figure 4). Contrarily, in urine from MPS I and MPS II patients treated with ERT, traces of DS and/or HS in GAG electrophoresis were still observed (Figure 3 and Figure 4).

## 4. Discussion

This study provides some novel insights regarding the positive impact of HSCT on bone growth of pediatric patients with both MPS I and MPS II after HSCT. We demonstrated that HSCT in patients with MPS I and MPS II resulted in the normalization of urinary GAG excretion and the absence of both dermatan and heparan sulfate (DS, HS) in GAG electrophoresis. We confirmed that hip dysplasia, kyphosis, and corneal clouding could worsen, aside from HSCT. 

First, we would like to highlight the importance of biochemical diagnosis of MPS. The identification of the stored substrate (GAGs) is crucial, and typically includes quantitative, semi-quantitative, and qualitative analysis (using electrophoresis or thin-layer chromatography) of GAGs excreted in urine. In our country, GAG electrophoresis is performed, allowing for the separation of GAGs with good sensitivity and reproducibility. In all our study patients (both MPS I and MPS II after HSCT), a biochemical improvement, defined as the normalization of urinary GAG excretion and the absence of both dermatan and heparan sulfate (DS, HS) in GAG electrophoresis, was observed. A dramatic decrease in DS in GAG electrophoresis in an MPS I patient was noted for the first time by Hobbs et al. several dozen years ago [10].

In our previous studies, we evaluated and described growth in untreated patients with MPS I, II, IVA, and VI [13,14,15]. We also published the results of an analysis of longitudinal anthropometric data of MPS II patients (*n* = 13) who started ERT before 6 years of age (range from 3 months to 6 years, mean 3.6 years, median 4 years) compared with a retrospective analysis of data for MPS II patients naïve to ERT (*n* = 50). The course of the average growth curve for MPS II patients treated with ERT was found to be very similar to the growth pattern in untreated patients—there was no statistically significant difference in the mean increase in body height between these two groups [16]. However, our results were not consistent with the study of Jones et al. [17] in the group of patients enrolled in HOS, the Hunter Outcome Survey. This study showed that the slope after treatment was significantly improved compared with before treatment [17]. Similarly to our observations in the group of MPS II patients treated with ERT, we did not observe a positive effect of recombinant human alpha-L-iduronidase (laronidase) on the growth of patients with MPS I [14]. Anthropometric features of 14 patients with MPS I were followed from birth until the introduction of ERT (group 1—in the 1st year of life, group 2—in the 3rd year of life), after 52–260 weeks of ERT, and periodically during treatment. After 96–260 weeks of ERT, patients receiving laronidase (group 1) did not show statistically significant improvement compared with group 2 [14].

In this study, MPS I and MPS II growth curves of patients after HSCT are compared to curves of untreated patients. Both MPS I and MPS II patients presented with height gain after HSCT compared to curves of untreated patients. This observation could be explained by the phenomenon of catch-up growth—that is, after removing the factor that limits the growth, the body tends to grow faster, making up for the deficiencies and equalizing its own reaction norm.

Up to now, only in the study of Cattoni et al. has a systematic comparison with the reference auxological data of untreated Hurler patients been demonstrated [18]. The authors observed that HSCT positively affected growth and provided transplanted patients with remarkable height gain. In all the remaining reports of MPS I, a comparison with the healthy population was performed. Long term, there was progressive deviation from the reference curves, with a large proportion of patients presenting with a final height below −2 SDS [19].

Regarding MPS II, Patel et al. assessed the impact of ERT and HSCT on growth in a group of 44 Japanese male patients with MPS II (26 patients had been treated with ERT, 12 patients had been treated with HSCT, and 6 had been treated with both ERT and HSCT) [20]. They found that MPS II patients who had been treated with either ERT or HSCT had increased height compared to untreated patients. Additionally, no significant difference in growth impact between patients treated with either ERT or HSCT from 4 to 12 years of age was observed. These observations are in line with the results of our study. All MPS II patients were treated with HSCT early in life (<2 years of age), and a height gain after HSCT was observed.

Bone disease (along with CNS impairment) constitutes the greatest therapeutic challenge in MPS due to limited penetration of ERT into poorly vascularized tissues (e.g., cartilage) and to the irreversibility of some lesions at the time of ERT/HSCT. Also, the therapeutic effect of HSCT in bone pathology is not very well understood. Santi et al. recently demonstrated in a mouse model of MPS I that BMT significantly reduced the widening of the long bones, but there was also complete normalization of long bone thickness [21]. Pievani et al. provided more detailed characteristics of bone markers, including trabecular number and separation, cortical thickness, and bone mineral volume [22]. This revealed significant differences between untreated and nBMT MPS I mice. All MPS I mice treated with BMT displayed above-mentioned bone parameter values comparable to those of wild-type mice, confirming significant improvements in the skeletal phenotype approaching complete normalization of each parameter tested.

The pathogenesis of growth impairment in patients with MPS can mostly be regarded as the effect of the pathological storage of dermatan sulfate (DS) in cartilage, bones, and growth plates. Simonaro et al. suggested that the main affected tissue is the cartilage rather than the bone itself. It has been also demonstrated that the accumulation of GAG exerts a detrimental effect on bone deposition by inducing dysfunctional osteoblastic activity [23]. Hinek and Wilson reported that elastogenesis takes place in the shaft of long bones during fetal life, and accumulation of DS by fibroblasts induces functional deficiency in the elastin-binding protein and, consequently, leads to disruption of normal elastogenesis [24]. DS damage to tropoelastin could be relevant to skeletal pathology in patients with MPS disease.

Based on our anthropologic (height gain after HSCT) and biochemical (absence of DS in GAG electrophoresis) results and the pathogenesis of growth impairment (chondrocytes damage) in MPS, we can assume that the absence of DS after HSCT could lead to normal bone length growth (depending on chondrocytes). On the other hand, HSCT has no impact on the growth of flat bones, which involves osteoblasts arising from mesenchymal tissue damaged in fetal life.

A limitation of our study is the small group of patients; however, the presented results shed light on the mechanism of compromised growth in patients with MPS, as well as the positive effect of HSCT. Further studies in a mouse model of MPS, including histological analysis of bone growth plates and studies on growth-related gene expression, are needed to evaluate our observations and provide stronger evidence (i.e., normal chondrocyte morphology and function and over-expression of genes related with bone growth in length). 

## 5. Conclusions

Both MPS I and MPS II patients presented with height gain after HSCT compared to the curves of untreated patients.

The absence of dermatan sulphate after HSCT could provide normal bone length growth.

## Figures and Tables

**Figure 1 diagnostics-14-01956-f001:**
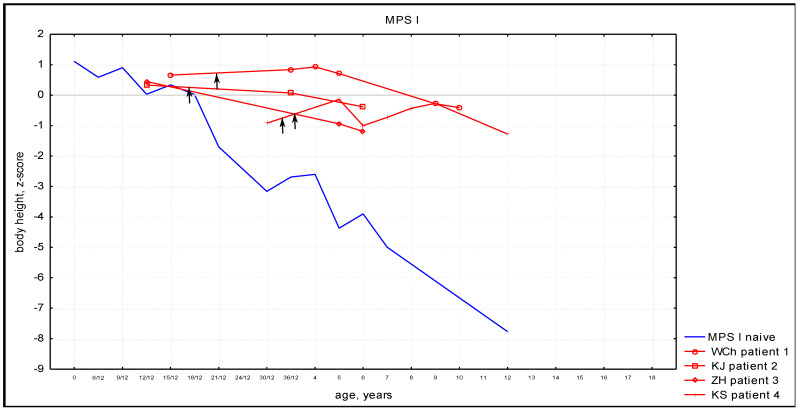
Body height of MPS I patients after HSCT (red lines; arrow—time of HSCT) with comparison with untreated patients (blue line).

**Figure 2 diagnostics-14-01956-f002:**
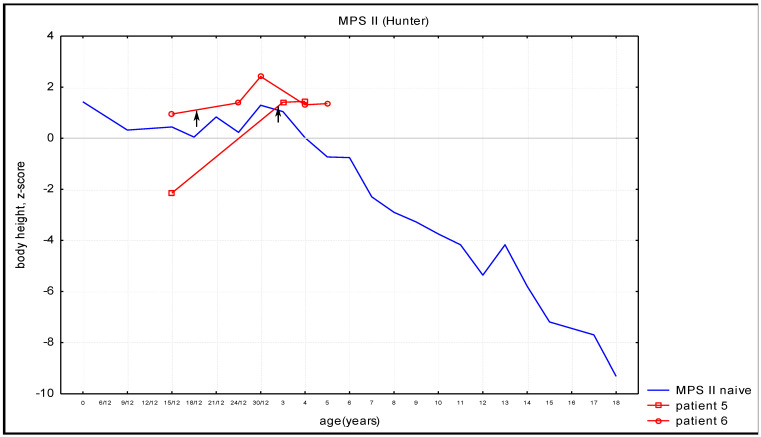
Body height of MPS II patients after HSCT (red lines; arrow—time of HSCT) with comparison with untreated patients (blue line).

**Figure 3 diagnostics-14-01956-f003:**
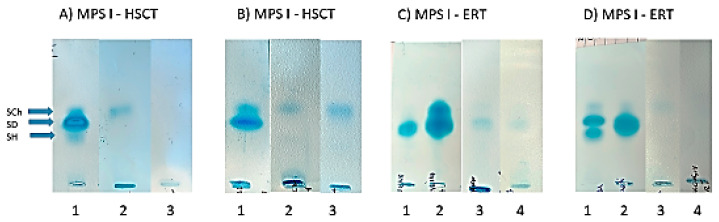
Changes in GAG electrophoresis in MPS I patients after HSCT and ERT. Abbreviations: SCh—chondroitin sulphate; SD—dermatan sulphate; SH—heparan sulphate; GAG—glycosaminoglycan; ERT—enzyme replacement therapy; HSCT—hematopoietic stem cell transplantation. (**A**) 1—before treatment; 2—after 2 years of HSCT; 3—after 4 years of HSCT. (**B**) 1—before treatment; 2—after 2 years of HSCT; 3—after 3 years of HSCT. (**C**) 1—before treatment; 2—control sample; 3—after 11 years of ERT; 4—after 21 years of ERT. (**D**) 1—control sample; 2—before treatment; 3—after 11 years of ERT; 4—after 21 years of ERT.

**Figure 4 diagnostics-14-01956-f004:**
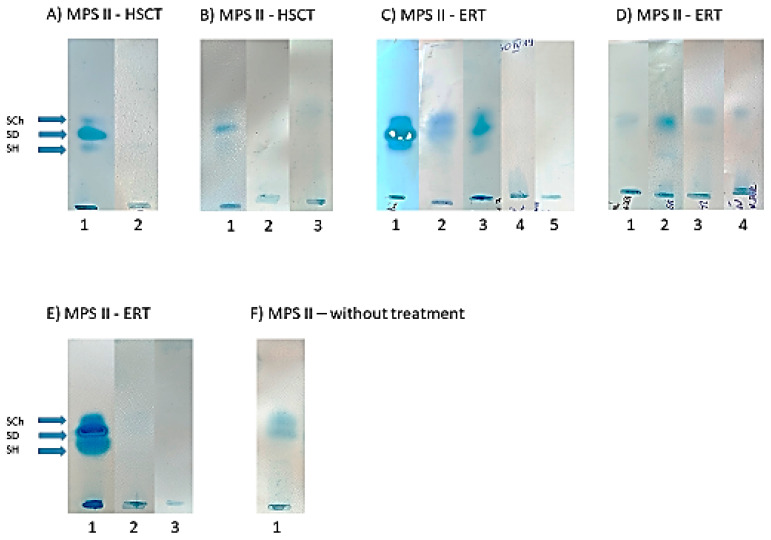
Changes in GAG electrophoresis in MPS II patients after HSCT and ERT. Abbreviations: SCh—chondroitin sulphate; SD—dermatan sulphate; SH—heparan sulphate; GAG—glycosaminoglycan; ERT—enzyme replacement therapy; HSCT—hematopoietic stem cell transplantation. (**A**) 1—before treatment; 2—after 9 months of HSCT. (**B**) 1—before treatment; 2—after 3 years of HSCT; 3—after 3 years and 4 months of HSCT. (**C**) 1—before treatment; 2—after 7.5 years of ERT; 3—after. 2.5 years of ERT; 4—after 16 years of ERT; 5—after 21 years of ERT. (**D**) 1—before treatment; 2—after 1.5 years of ERT; 3—after 3 years of ERT; 4—after 4.5 years of ERT. (**E**) 1—before treatment; 2—after 6 years of HSCT; 3—after 10 years of HSCT. (**F**) 1—MPS II without treatment.

**Table 1 diagnostics-14-01956-t001:** Clinical, biochemical, and molecular data of the study patients (Abbreviations: DBS—dried blood spot; GAGs—glycosaminoglycans; DS—dermatan sulphate; HS—heparan sulphate; y—year; m—month; pc—percentile).

Patient No. Type of MPS Genotype	Age	Anthropometric Analyses	Alpha-Iduronidase (MPS I) or Iduronate 2-Sulfatase (MPS II) Activity in DBS	Alpha-Iduronidase (MPS I) or Iduronate 2-Sulfatase (MPS II) Activity in Leukocytes	GAGs in Urine—Quantitative Analysis with DMB	GAGs in Urine—Semi-Quantitative Analysis with Cetylpyridinium Chloride	GAGs Electrophoresis	Clinical Outcome
1 MPS I c.208C > T, p.(Gln70*)/ c.208C > T, p.(Gln70*)	13 m	Weight 10.9 kg (50 pc) Height 79.5 cm (50 pc)	<1.1 (limit of quantification) µmol/L/h (ref. > 3.0 µmol/L/h)	10 nmol/mg protein/18 h (129 ± 40.5)	158.7 (4.21–18.7) mg/mmol creatinine)	699 (151 ± 77) mg/g creatinine	DS and subtle HS	inguinal hernia, subtle corneal clouding, thickened mitral valve, mild liver enlargement, limited range of motion in the glenohumeral joints
	5.5 y (3 y after HSCT)	Weight 21.2 kg (50 pc) Height 113.8 cm (35 pc)	1.7 µmol/L/h (ref. > 3.0 µmol/L/h)	180 nmol/mg protein/18 h (129 ± 40.5)	17.6 (4.21–18.7 mg/mmol creatinine)	55 (113 ± 46) mg/g creatinine	Normal results	stable thickening of mitral valve, corneal opacity over the entire surface, qualification for adenotomy, normal cognitive development, stable limitation of the range of motion in the glenohumeral joints, narrowing of the spinal canal (5.5 y)
2 MPS I c.208C > T, p.(Gln70*)/ c.208C > T, p.(Gln70*)	14 m	Weight 11.6 kg (50–75 pc) Height 80.5 cm (50–75 pc)	<1.1 (limit of quantification) µmol/L/h (ref. > 3.0 µmol/L/h)	9.72 nmol/mg protein/18 h (129 ± 40.5)	137 (4.21–18.7) mg/mmol creatinine)	648 (151 ± 77) mg/g creatinine	DS and subtle HS	bilateral hearing loss, thickened mitral valve, mild liver enlargement, limited range of motion in the glenohumeral joints
	5.5 y	Weight 20.3 kg (19 pc) Height 109.2 cm (12 pc)	5.9 µmol/L/h (ref. > 3.0 µmol/L/h)	178 nmol/mg protein/18 h (129 ± 40.5)	16.4 (4.21–18.7 mg/mmol creatinine)	55 (113 ± 46) mg/g creatinine	Traces of DS and HS	mild hearing loss, carpal tunnel syndrome diagnosed at 5 y, normal cognitive development, stable limitation of the range of motion in the glenohumeral joints, carpal tunnel syndromne (5 y)
3 MPS I c.208C > T, p.(Gln70*)/ c.208C > T, p.(Gln70*)	13 m	Weight 10.3 kg (10–25 pc) Height 80.6 cm (75–90 pc)	<1.1 (limit of quantification) µmol/L/h (ref. > 3.0 µmol/L/h)	0.01 nmol/mg protein/18 h (129 ± 40.5)	253 (4.21–18.7) mg/mmol creatinine)	1603 (151 ± 77) mg/g creatinine	DS	Thickened mitral valve, bilateral hearing loss, corneal clouding, mild liver enlargement, limited range of motion in the glenohumeral joints
	12 y	Weight 40.2 kg (77 pc) Height 137.2 cm (34 pc)	3.9 µmol/L/h (ref. > 3.0 µmol/L/h)	87 nmol/mg protein/18 h (129 ± 40.5)	15.3 (4.21–18.7 mg/mmol creatinine)	66 (115 ± 61) mg/g creatinine	Normal results	no carpal tunnel syndrome, progressive thoracic kyphosis, bilateral hip dysplasia, normal intellectual development, stable thickening of mitral valve
4 MPS I c.208C > T, p.(Gln70*)/ c.208C > T, p.(Gln70*)	12 y (age at diagnosis: 16 months)	Weight 49.7 kg (78 pc) Height 144.3 cm (10 pc)	4.1 µmol/L/h (ref. > 3.0 µmol/L/h)	132 nmol/mg protein/18 h (129 ± 40.5)	9.6 (4.21–18.7 mg/mmol creatinine)	134 (115 ± 61) mg/g creatinine	Normal results	Bilateral hip dysplasia, progressive lumbar lordosis, mild hearing loss, corneal opacity over the entire surface, carpal tunnel syndrome (12 y), normal intellectual development
5 MPS II c.1134_1152dup, p.(Asp385ProfsX7)	14 m	Weight 15.3 kg (90–97 pc) Height 86.5 cm (>97 pc)	<0.8 (limit of detection) µmol/L/h (ref. > 5.6 µmol/L/h)	2.8 nmol/mg protein/4 h (354 ± 85.9)	111.7 (9.52–26.9 mg/mmol creatinine)	944 (151 ± 77) mg/g creatinine	DS, HS	limited range of motion in the glenohumeral joints, qualified for adenectomy, mild liver enlargement, thickened mitral valve
	5 y (3 y after HSCT)	Weight 20.1 kg (62 pc) Height 116.2 cm (91 pc)	3.1 µmol/L/h (ref. > 5.6 µmol/L/h)	28.8 nmol/mg protein/4 h (354 ± 85.9)	13.2 (9.52–26.9 mg/mmol creatinine)	207 (151 ± 77) mg/g creatinine	Normal results	normal intellectual development, stable limitation of the range of motion in the glenohumeral joints, stable thickening of the mitral valve
6 MPS II IDS, loss of exons 01 to 08	14 m	Weight 15.3 kg (90–97 pc) Height 86.5 cm (>97 pc)	<2.8 (limit of quantification) µmol/L/h (ref. > 5.6 µmol/L/h)	2.8 nmol/mg protein/4 h (354 ± 85.9)	103.7 (9.52–26.9 mg/mmol creatinine)	957 (151 ± 77) mg/g creatinine	DS, HS	moderate hearing loss, qualified for adenectomy, unilateral hip dysplasia, limited range of motion in the glenohumeral and hip joints, mild liver enlargement, thickened mitral valve
	3 y 2 m (1 y and 3 m after HSCT)	Weight 22.3 kg (>97 pc) Height 102.8 cm (92 pc)	4.2 µmol/L/h (ref. > 3.0 µmol/L/h)	35.9 nmol/mg protein/4 h (354 ± 85.9)	13.2 (9.52–26.9 mg/mmol creatinine)	226 (151 ± 77) mg/g creatinine	Normal results	moderate hearing loss, hearing aids, stable limitation of the range of motion in the glenohumeral and hip joints, stable thickening of mitral valve

## Data Availability

The original contributions presented in the study are included in the article/Appendix A, further inquiries can be directed to the corresponding author/s.

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
