# Peer review of "Body Height of MPS I and II Patients after Hematopoietic Stem Cell Transplantation: The Impact of Dermatan Sulphate"

_diagnostics, 2024, doi:10.3390/diagnostics14171956_

Round 1
Reviewer 1 Report
Comments and Suggestions for Authors
Dear Authors,
Thank you for submitting your research paper on the height development of mucopolysaccharidosis (MPS) patients following hematopoietic stem cell transplantation (HSCT). Your study provides valuable insights into this important area. After careful review, I believe the manuscript has potential for publication, but requires some significant revisions. Here are my suggestions:
Major Revisions:
1. Sample size: Consider increasing the sample size or thoroughly discuss the limitations of the small sample. Explore the possibility of multi-center collaboration.
2. Control group: Include MPS patients receiving enzyme replacement therapy (ERT) as a control group to more comprehensively evaluate the effects of HSCT.
3. Mechanism exploration: Deepen the investigation of potential mechanisms, such as histological analysis of bone growth plates or studies on growth-related gene expression.
Minor Revisions:
1. Graph presentation: Present individual patient growth curves alongside normal population growth curves in the same graph for better comparison.
2. Methodological description: Provide detailed information on the HSCT protocol, including conditioning regimens and donor sources.
3. Discussion expansion: Compare your results with existing literature and discuss potential clinical applications.
4. Height measurement standardization: Clarify how you addressed the potential impact of skeletal deformities on height measurements in MPS patients.
5. Clinical features supplementation: Provide more clinical information about the patients, such as cognitive function and organ involvement.
I believe these revisions will make your study more comprehensive and rigorous, providing stronger evidence for HSCT treatment in MPS patients. I look forward to receiving your revised manuscript. Please don't hesitate to contact me if you have any questions.
Sincerely,
Minor editing of English language required.
Author Response
Reviewer 1
Dear Authors,
Thank you for submitting your research paper on the height development of mucopolysaccharidosis (MPS) patients following hematopoietic stem cell transplantation (HSCT). Your study provides valuable insights into this important area. After careful review, I believe the manuscript has potential for publication, but requires some significant revisions. Here are my suggestions.
Answer: Dear Reviewer! We are very grateful for Your valuable comments.
Major Revisions:
- Sample size: Consider increasing the sample size or thoroughly discuss the limitations of the small sample. Explore the possibility of multi-center collaboration.
2. Control group: Include MPS patients receiving enzyme replacement therapy (ERT) as a control group to more comprehensively evaluate the effects of HSCT.
Answer: Of course, the comparison with MPS patients treated with ERT, will be very important, also from methodological point of view. However, the studies on growth in MPS patients, both untreated and treated with ERT, were previously published (by our group) – see below. Thus, we discussed the limitations of the small sample.
The following part was added to the discussion: In our previous studies, we evaluated and described growth in untreated patients with MPS I, II, IVA and VI [13-15]. We have also published the results of analysis of longitudinal anthropometric data of MPS II patients (n = 13) who started ERT before 6 years of age (range from 3 months to 6 years, mean 3.6 years, median 4 years) compared with retrospective analysis of data for MPS II patients naïve to ERT (n = 50). The course of the average growth curve for MPS II patients treated with ERT was found very similar to growth pattern in untreated patients – there was no statistically significant difference in the mean increase in body height between these two groups [16]. However, our results were not consistent with the study of Jones et al. [17] in the group of patients enrolled in HOS – the Hunter Outcome Survey. This study showed that the slope after treatment was significantly improved compared with before treatment [17]. Similarly to our observations in the group of MPS II patients treated with ERT, we did not observe the positive effect of recombinant human alpha-L-iduronidase (laronidase) on growth of patients with MPS I [18]. Anthropometric features of 14 patients with MPS I were followed from birth until the introduction of ERT (group 1 – in the 1st year of life, group 2 – in the 3rd year of life), after 52-260 weeks of ERT and periodically during treatment. After 96-260 weeks of ERT, patients receiving laronidase (group 1) compared with group 2 did not show statistically significant improvement [18].
The limitation of our study is the small group of patients, however the presented results shed a light on the mechanism of a compromised growth in patients with MPS, and the positive effect of HSCT. Further studies, including histological analysis of bone growth plates and studies on growth-related gene expression are needed to evaluate our ob-servations and provide stronger evidence.
Mechanism exploration: Deepen the investigation of potential mechanisms, such as histological analysis of bone growth plates or studies on growth-related gene expression.
Answer: The pathogenesis of growth impairment in patients with MPS can be mostly re-garded as the effect of the pathological storage of dermatan sulfate (DS) in cartilages, bone, and growth plate. Simonaro et al. suggested that the main affected tissue is the cartilage rather than the bone itself. It has been also demonstrated that the accumulation of GAG plays a detrimental effect on bone deposition by inducing a dysfunctional osteoblastic activity [22]. Hinek and Wilson reported that elastogenesis takes place in the shaft of long bones during foetal life, and accumulation of DS by fibroblasts induces the functional deficiency in the elastin-binding protein and, consequently, leads to disruption of normal elastogenesis [23]. DS damage to tropoelastin could be relevant to skeletal pathology found in MPS disease.
Based on our anthropologic results and pathogenesis of growth impairment in MPS, we could assume that the absence of DS after HSCT provides normal bone growth in length which is depended on chondrocytes. On the other hand, HSCT has no impact of the growth of flat bones, which involves osteoblasts arising from mesenchymal tissue damaged in fetal life.
The limitation of our study is the small group of patients, however the presented results shed a light on the mechanism of a compromised growth in patients with MPS, and the positive effect of HSCT. Further studies, including histological analysis of bone growth plates and studies on growth-related gene expression are needed to evaluate our observations and provide stronger evidence.
Minor Revisions:
1. Graph presentation: Present individual patient growth curves alongside normal population growth curves in the same graph for better comparison.
Answer: Figures 1 and 2 described the z-score of body height of MPS II patients after HSCT (red lines; arrow – time of HSCT) with comparison with un-treated patients (blue line). Individual patient’s growth charts were provided as Supplementary Figure 1.
Methodological description: Provide detailed information on the HSCT protocol, including conditioning regimens and donor sources.
Answer: The following part was added to the main text of manuscript: Transplant procedures were performed according to recommendations of Inborn Errors Working Party European Society for Blood and Marrow Transplantation (EBMT). Condi-tioning regimen was myeloablative and consisted of busulfan or treosulfan combined with fludarabine. Stem cells were obtained from unstimulated bone marrow of unrelated donors. Antitymocyte globuline, cyclosporin and mycofenolete mofetil were used as graft versus host disease prophylaxis.
Discussion expansion: Compare your results with existing literature and discuss potential clinical applications .
Answer: The comparison was provided in main text of manuscript.
The following part was also added into discussion: We would like to highlight the importance of biochemical diagnosis of MPS. The identification of the stored substrate (GAG) is crucial, which typically includes both the quantitative and semi-quantitative and also qualitative analysis (using electrophoresis or thin-layer chromatography) of GAG excreted in urine. In our country, GAG electrophoresis is performed allowing separation of GAG with good sensitivity and reproducibility.
Height measurement standardization: Clarify how you addressed the potential impact of skeletal deformities on height measurements in MPS patients.
Answer: The study group patients did not present with skeletal deformities. Also, HSCT were performed in early stages of patients’ life (median age of 2 years; range 18 months – 2 years and 6 months) thus the skeletal deformities did not impact the patients’ growth.
Patients with large skeletal deformities are generally measured in a lying position along the curves of the body with a centimeter.
Clinical features supplementation: Provide more clinical information about the patients, such as cognitive function and organ involvement.
Answer: Clinical outcome was provided in the corrected version of Table 1.

Reviewer 2 Report
Comments and Suggestions for Authors
I read with interest the work of P. Lipinki et al:
Body height of MPS I and II patients after hematopoietic stem cell transplantation: the impact of dermatan sulphate
The work seemed interesting, concise and substantially well written.
Abstract, materials and methods section, the authors indicated the type of study (observational single center) instead of the patients evaluated and the determinations performed.
Materials and Methods: line 91-94. I would think it would be useful if the authors provided further details regarding the transplant procedures, for example whether the donor was family or (as I believe) unrelated. Patient 2, in fact, showed a late failure to engraft with the development of hemolytic anemia. The transplant presented a major ABO incompatibility, how was it prepared for the transplant?
Figures 1 and 2 are not interpretable. It will be necessary to redo them by preparing the differences in traits (colors?) between the various patients.
Comments on the Quality of English Language
English language needs some revisions in example:
line 40: erase "undoubtedly
line 46: use recognized intead of distinguished
line 113: groups of patients
Author Response
Reviewer 2
I read with interest the work of P. Lipinki et al:
Body height of MPS I and II patients after hematopoietic stem cell transplantation: the impact of dermatan sulphate
The work seemed interesting, concise and substantially well written.
Answer: Dear Reviewer! We are very grateful for Your valuable comments. Please, find attached the manuscript corrected as advised.
Abstract, materials and methods section, the authors indicated the type of study (observational single center) instead of the patients evaluated and the determinations performed.
Answer: Of course, the comparison with MPS patients treated with ERT, will be very important, also from methodological point of view. However, the studies on growth in MPS patients, both untreated and treated with ERT, were previously published (by our group) – see below. Thus, we discussed the limitations of the small sample.
The following part was added to the discussion: In our previous studies, we evaluated and described growth in untreated patients with MPS I, II, IVA and VI [13-15]. We have also published the results of analysis of longitudinal anthropometric data of MPS II patients (n = 13) who started ERT before 6 years of age (range from 3 months to 6 years, mean 3.6 years, median 4 years) compared with retrospective analysis of data for MPS II patients naïve to ERT (n = 50). The course of the average growth curve for MPS II patients treated with ERT was found very similar to growth pattern in untreated patients – there was no statistically significant difference in the mean increase in body height between these two groups [16]. However, our results were not consistent with the study of Jones et al. [17] in the group of patients enrolled in HOS – the Hunter Outcome Survey. This study showed that the slope after treatment was significantly improved compared with before treatment [17]. Similarly to our observations in the group of MPS II patients treated with ERT, we did not observe the positive effect of recombinant human alpha-L-iduronidase (laronidase) on growth of patients with MPS I [18]. Anthropometric features of 14 patients with MPS I were followed from birth until the introduction of ERT (group 1 – in the 1st year of life, group 2 – in the 3rd year of life), after 52-260 weeks of ERT and periodically during treatment. After 96-260 weeks of ERT, patients receiving laronidase (group 1) compared with group 2 did not show statistically significant improvement [18].
The limitation of our study is the small group of patients, however the presented results shed a light on the mechanism of a compromised growth in patients with MPS, and the positive effect of HSCT. Further studies, including histological analysis of bone growth plates and studies on growth-related gene expression are needed to evaluate our ob-servations and provide stronger evidence.
Materials and Methods: line 91-94. I would think it would be useful if the authors provided further details regarding the transplant procedures, for example whether the donor was family or (as I believe) unrelated. Patient 2, in fact, showed a late failure to engraft with the development of hemolytic anemia. The transplant presented a major ABO incompatibility, how was it prepared for the transplant?
Answer: The following part was added to the main text of manuscript: Transplant procedures were performed according to recommendations of Inborn Errors Working Party European Society for Blood and Marrow Transplantation (EBMT). Condi-tioning regimen was myeloablative and consisted of busulfan or treosulfan combined with fludarabine. Stem cells were obtained from unstimulated bone marrow of unrelated donors. Antitymocyte globuline, cyclosporin and mycofenolete mofetil were used as graft versus host disease prophylaxis.
Regarding the patient 2, this is the exact data of HSCT:
1st HSCT – BM MUD – Treosulfan + Fludarabine
2nd HSCT – BM MUD - Busulfan + Fludarabine
Figures 1 and 2 are not interpretable. It will be necessary to redo them by preparing the differences in traits (colors?) between the various patients.
Answer: Both Figures were corrected.
Figure 1 - Body height of MPS I patients after HSCT (red lines; arrow – time of HSCT) with comparison with un-treated patients (blue line).
Figure 2 - Figure 2. Body height of MPS II patients after HSCT (red lines; arrow – time of HSCT) with comparison with un-treated patients (blue line).

Round 2
Reviewer 1 Report
Comments and Suggestions for Authors
Dear Authors,
Thank you for your thorough responses to my previous review comments. Your revisions have significantly improved the quality of the manuscript. After carefully reviewing your replies and revisions, I have a few minor suggestions for further improvement:
1. Regarding the histological analysis of growth plates and studies on growth-related gene expression, you mention these as directions for future research in the discussion section. I suggest expanding on this slightly in the methods section or future outlook, briefly explaining how these studies might be conducted and what insights you expect to gain.
2. You have provided rich clinical data in Table 1, which is excellent. However, I suggest providing a brief summary of this data in the results section, highlighting some key clinical findings or trends.
3. Concerning the impact of skeletal deformities on height measurements, you explained that the study group patients did not have significant deformities. I recommend explicitly stating this in the methods section and briefly describing how you determined that patients did not have deformities that would affect measurements.
4. In discussing the pathological mechanisms of growth impairment in MPS, you provide good background information. I suggest further elaborating on how these mechanisms specifically relate to your observations of improved growth post-HSCT.
5. The individual patient growth charts in Supplementary Figure 1 are very helpful. I recommend briefly mentioning these individual differences in the main text and briefly discussing possible reasons for the variations.
Once again, thank you for your hard work and commitment to improving the manuscript. I believe these minor revisions will further enhance the value of your research. I look forward to seeing the revised version.
Sincerely,
Reviewer
Comments on the Quality of English LanguageMinor editing of English language required.
Author Response
Dear Reviewer
We are very grateful for Your comments. We hope the revised manuscript will be worth publication.
- Regarding the histological analysis of growth plates and studies on growth-related gene expression, you mention these as directions for future research in the discussion section. I suggest expanding on this slightly in the methods section or future outlook, briefly explaining how these studies might be conducted and what insights you expect to gain.
Answer: Further studies in a mouse model of MPS, including histological analysis of bone growth plates and studies on growth-related gene expression are needed to evaluate our observations and provide stronger evidence (i.e. normal chondrocytes morphology and function, over-expression of genes related with bone growth in length).
- You have provided rich clinical data in Table 1, which is excellent. However, I suggest providing a brief summary of this data in the results section, highlighting some key clinical findings or trends.
Answer: 3.2 Clinical outcome
Thoracolumbar spine kyphosis (MPS I) as well as hip dysplasia (MPS I and II) were observed in all patients at the time of diagnosis. There was observed a tendency to progression of both spine kyphosis and hip dysplasia, despite HSCT. However, only 1 patient (Pt 6) required surgical intervention (due to hip luxation). At diagnosis, there was observed the biggest restriction in passive range of movement (ROM) in shoulder abduction and flexion. After HSCT, there was observed a stabilization or slight improvement.
No patient had hydrocephalus. Narrowing of the spinal canal was observed in one patient (Pt 1) starting at age of 5.5 years.
No patient presented with carpal tunnel syndrome before HSCT. Two patients (Pt 2 and 4) developed carpal tunnel syndrome, at 5 and 12 years, respectively.
Corneal clouding was observed in all patients at the time of diagnosis of MPS I. Despite HSCT, corneal clouding worsened in all patients.
All patients but one (Pt 1) underwent adenectomy with tympanostomy tubes fitting before HSCT. The latter was qualified for these procedures after HSCT at 5.5 years of age.
Bilateral sensorineural hearing impairment was observed in 4 patients and was assessed as mild or moderate (40-55 dB) at the time of diagnosis. Long-term hearing appeared to improve in two patients (unilateral mild hearing loss) and stabilize in two other patient after HSCT.
Mitral valve thickening was observed in all patients before HSCT and remained stable after HSCT.
All the patient presented with normal intellectual development after HSCT.
A detailed clinical characteristics of the study patients was provided in Table 1.
- Concerning the impact of skeletal deformities on height measurements, you explained that the study group patients did not have significant deformities. I recommend explicitly stating this in the methods section and briefly describing how you determined that patients did not have deformities that would affect measurements.
Answer: The study group patients did not present with skeletal deformities that would affect measurements. Also, the comparable measurements in a lying position along the curves of the body with a centimeter confirmed the above-mentioned.
- In discussing the pathological mechanisms of growth impairment in MPS, you provide good background information. I suggest further elaborating on how these mechanisms specifically relate to your observations of improved growth post-HSCT.
Bone disease (along with CNS impairment) constitutes the greatest therapeutic challenge in MPS due to limited penetration of ERT into poorly vascularized tissues (e.g., cartilage) and to the irreversibility of some lesions at the time of ERT/HSCT. Also, the therapeutic effect of HSCT in bone pathology is not very well understood. Santi et al. have recently demonstrated in a mouse model of MPS I that BMT significantly reduced widening of the long bones but also complete normalization of long bone thickness [22]. Pievani et al. provided a more detailed characteristics on bone markers, including trabecular number and separation, cortical thickness, and bone mineral volume [23]. This re- vealed significant differences between untreated and nBMT MPS I mice. All MPS I mice treated with BMT displayed the above-mentioned bone parameter values comparable to those of wild type mice, confirming the significant improvements in skeletal phenotype approaching complete normalization of each parameter tested.
The pathogenesis of growth impairment in patients with MPS can be mostly regarded as the effect of the pathological storage of dermatan sulfate (DS) in cartilages, bone, and growth plate. Simonaro et al. suggested that the main affected tissue is the cartilage rather than the bone itself. It has been also demonstrated that the accumulation of GAG plays a detrimental effect on bone deposition by inducing a dysfunctional osteoblastic activity [24]. Hinek and Wilson reported that elastogenesis takes place in the shaft of long bones during foetal life, and accumulation of DS by fibroblasts induces the functional deficiency in the elastin-binding protein and, consequently, leads to disruption of normal elastogenesis [25]. DS damage to tropoelastin could be relevant to skeletal pathology found in MPS disease.
Based on our anthropologic (height gain after HSCT) and biochemical (absence of DS in GAG electrophoresis) results and pathogenesis of growth impairment (chondrocytes damage) in MPS, we could assume that the absence of DS after HSCT could provide normal bone growth in length (depending on chondrocytes). On the other hand, HSCT has no impact of the growth of flat bones, which involves osteoblasts arising from mesenchymal tissue damaged in fetal life.
Answer:
- The individual patient growth charts in Supplementary Figure 1 are very helpful. I recommend briefly mentioning these individual differences in the main text and briefly discussing possible reasons for the variations.
Answer: Individual patient’s growth charts were provided as Supplementary Figure 1. All the patients presented with a height gain after HSCT.